# Effects of the Electrical Properties of SnO_2_ and C60 on the Carrier Transport Characteristics of p-i-n-Structured Semitransparent Perovskite Solar Cells

**DOI:** 10.3390/nano13243091

**Published:** 2023-12-06

**Authors:** Hoang Minh Pham, Syed Dildar Haider Naqvi, Huyen Tran, Hung Van Tran, Jonabelle Delda, Sungjun Hong, Inyoung Jeong, Jihye Gwak, SeJin Ahn

**Affiliations:** 1Photovoltaics Research Department, Korea Institute of Energy Research, Daejeon 34129, Republic of Koreajon@kier.re.kr (J.D.); iy1024@kier.re.kr (I.J.);; 2Department of Renewable Energy Engineering, University of Science and Technology, Daejeon 34113, Republic of Korea

**Keywords:** transparent perovskite solar cells, SnO_2_, C60, electrical properties, simulation

## Abstract

Recently, metal halide perovskite-based top cells have shown significant potential for use in inexpensive and high-performance tandem solar cells. In state-of-the-art p-i-n perovskite/Si tandem devices, atomic-layer-deposited SnO_2_ has been widely used as a buffer layer in the top cells because it enables conformal, pinhole-free, and highly transparent buffer layer formation. In this work, the effects of various electrical properties of SnO_2_ and C60 layers on the carrier transport characteristics and the performance of the final devices were investigated using a numerical simulation method, which was established based on real experimental data to increase the validity of the model. It was found that the band alignment at the SnO_2_/C60 interface does, indeed, have a significant impact on the electron transport. In addition, as a general design rule, it was suggested that at first, the conduction band offset (CBO) between C60 and SnO_2_ should be chosen so as not to be too negative. However, even in a case in which this CBO condition is not met, we would still have the means to improve the electron transport characteristics by increasing the doping density of at least one of the two layers of C60 and/or SnO_2_, which would enhance the built-in potential across the perovskite layer and the electron extraction at the C60/SnO_2_ interface.

## 1. Introduction

In recent years, metal halide perovskite solar cells (MHPSCs) have attracted great attention as a novel light-absorbing material for use in solar cells [1]. The PCE (power conversion efficiency) for single-junction devices increased significantly, from 3.8%, in the first report by Miyasaka’s group in 2009 [2], to 26.1% [3], which places them among the most favorable candidates for opening avenues in the field of SCs. This great performance originates from the excellent optical and electronic properties of perovskite [4], including high absorption coefficients, long diffusion lengths, high mobility of the charge carriers [5,6,7], higher defect tolerance with low trap density [8,9], tunable bandgaps [10], and simple, low-cost, and low-temperature fabrication routes [11,12]. These unique properties of MHPSCs also render them attractive candidates as top cells for application in low-cost and high-efficiency tandem solar cells.

In general, the structure of MHPSCs is classified into two categories, n-i-p and p-i-n structures, depending on the sequence of deposition of each functional layer. Although n-i-p-structured MHPSCs have been reported to show superior performances to p-i-n-structured ones in a single-junction configuration so far, p-i-n devices have recently been used more prominently in monolithic tandem solar cells due to several of their unique advantages over n-i-p devices, including their substantially low-temperature processes (no need for a high-temperature sintering step for the mesoporous TiO_2_ electron transport layer), higher matching current density (no need to use Spiro-MeOTAD hole transport materials with high parasitic absorption), and structurally better compatibility with various bottom cells, such as crystalline silicon (c-Si) and thin film solar cells (Cu(In,Ga)Se_2_ and Cu_2_ZnSn(Se,S)_4_).

A typical monolithic perovskite/silicon tandem device utilizing p-i-n MHPSCs as the top cells is a Si bottom cell/interconnection layer/hole-transporting layer (HTL)/perovskite/electron-transporting layer (ETL)/buffer layers/transparent conducting oxide (TCO). In state-of-the-art devices, self-assembled monolayers (SAMs) such as Me-PACz, C60, SnO_2_, and ITO have reportedly been used as the HTL, ETL, buffer layer, and top TCO layers, respectively [13]. The role of an interconnection layer is to induce carrier recombination or tunneling, by which the total voltage output of the tandem device becomes the sum of the photovoltages of each sub cell [14]. Recently, tunneling recombination layers based on hydrogenated nanocrystalline Si (nc-Si:H)(p^+^)/(nc-Si:H)(n^+^) have been reported to fabricate high-efficiency MHPSC/c-Si tandem solar cells [15]. In the tandem cells, the primitive role of the buffer layer is to act as a physical barrier to prevent the C60 ETL layers from sputtering damage due to the ITO process. In addition to this role of the physical barrier, the buffer layer should also satisfy several requirements: proper band alignment in between the ETL and top TCO to facilitate electron transport to the top TCO, and enabling high sub-bandgap optical transmission to increase the photocurrent and damage-free nature of its process. Although there are a variety of n-type metal oxide materials as candidates for the buffer layer in p-i-n-structured top cells (TiO_x_, Nb_2_O_x_, and In_2_O_x_), SnO_2_ is the most prevailing material because of its high transparency, decent electrical properties, and chemical stability.

Among the various SnO_2_ fabrication methods, atomic layer deposition (ALD) has become the most frequently used one [16] because it enables a conformal, pinhole-free, and highly transparent buffer layer formation on rather rough surfaces of the underlying structures [17]. The high transparency is particularly important in the state-of-the art p-i-n perovskite/Si tandem devices, where the light is incident on the ETL side [18].

Because the primitive role of the buffer layer is to act as a physical barrier, most of the pioneering research has primarily focused on the optimization of ALD parameters with the aim of obtaining conformal and pinhole-free SnO_2_ layers without damaging the bottom layers [18]. However, as mentioned above, the electrical properties of SnO_2_ (in terms of both its inherent characteristics and matching with adjacent layers), and hence those of the SnO_2_/C60 interface, can have significant effects on the performance of the tandem device, which has been largely unexplored and was even ignored during ALD-based SnO_2_ optimization. Therefore, understanding the effect of band alignment at the SnO_2_/C60 interface on the device’s performance is an important topic in terms of the optimum choice of material properties, and will be the main theme of this study. It is well known that the process parameters of ALD can have profound effects on the optoelectrical properties of SnO_2_, including the optical bandgap, band positions, and carrier concentration. Those of C60 can even be mutually affected during the SnO_2_ process, i.e., thermally activated diffusion of constituent ions from the underlying perovskite. All of these may have a significant impact on the band alignment, particularly at C60/SnO_2_ interfaces, and determine the electron transport to the top ITO electrode. However, a fundamental understanding of this portion and the corresponding design rules for the optimization of the ALD process are still lacking.

In this work, we performed theoretical analyses aiming to understand the role of the electrical properties of SnO_2_ and C60 on the charge transport characteristic at the C60/SnO_2_ interface and the final device performance of p-i-n-structured perovskite top cells using a device simulator, the Solar Cell Capacitance Simulator (SCAPS), developed by the University of Ghent. In particular, the electron affinity of SnO_2_ and the doping concentration of both SnO_2_ and C60 were chosen as major variables because of their practical importance in determining the band alignment at the corresponding interface, as well as the fact that they are reported to vary significantly by experimental ALD conditions. The distinguished aspect of the simulation performed in this work is that most of the input parameters in the baseline models are extracted from real experimental data from both the opaque and semitransparent devices, which increases the reliability of our results. Otherwise, the simulation result may lead us to unrealistic and meaningless interpretations.

## 2. Materials and Methods

### 2.1. Materials

All the materials for the perovskite precursor and for evaporation were used as bought, without further purification. The lead iodide and lead bromide were purchased from TCI (Tokyo, Japan, 99.99%, trace metal basis) or Sigma-Aldrich (Burlington, MA, USA, 99.999%, trace metal basis). The FAI and MABr were purchased from Greatcell Solar Materials (Queanbeyan, NSW, Australia, grade > 99.88%), and the CsI was obtained from Alfa Aesar (Haverhill, MA, USA, 99.999%, trace metal basis). The Me-4PACz self-assembly monolayers (SAMs) were purchased from TCI. All the anhydrous solvents, such as DMF, DMSO, toluene, and antisolvents, as well as SAM solvents, such as ethanol, were obtained from Sigma-Aldrich. The C60 (99.99%) and BCP (>99.0%) were bought from Nano-C (Westwood, MA, USA) or TCI.

### 2.2. Device Fabrication and Characterization

#### 2.2.1. Opaque Device

The etched FTO glass substrates were subjected to 15 min of sonication in deionized water, acetone, and ethyl alcohol baths, in that sequence. The FTO substrates were then dried in an oven at 70 °C and treated with plasma ozone for 10 min to remove any organic residue from the surface and activate the surface for SAM application. Afterward, the substrates were transferred into a nitrogen-filled glovebox as an inert processing environment. The SAM was applied using a 2 mM solution of Me-4PACz via spin-coating at 3000 rpm for 30 s and annealing at 100 °C for 10 min. After the substrates were cooled to room temperature, they were washed by applying 150 µL ethanol during the spinning process 2 times. Then, 140 µL of a 1.4 M solution of Cs_0.05_FA_0.86_MA_0.09_Pb(I_0.91_Br_0.09_) [19] was dissolved in DMF: DMSO (4: 1 by vol) via stirring for 3 h, and then was poured and spread onto the FTO surface after the SAM was applied. The substrate was spun at 2000 rpm for 10 s and 4000 rpm for 30 s. A total of 200 µL of chlorobenzene was poured as an antisolvent via a cut pipette tip in the middle of the spinning substrate 10 s prior to the end of the spinning program, and then it was subjected to annealing at 100 °C for 20 min. The samples were then loaded into the vacuum chamber of a thermal evaporator, where 25 nm of C60 and 5 nm of BCP were sequentially deposited via thermal evaporation under a vacuum of <5 × 10^−6^ Torr. Finally, using a shadow mask with a contact area of 0.143 cm^2^, a silver counter electrode 95 nm thick was deposited via thermal evaporation under a high-vacuum condition of <5 × 10^−6^ Torr.

#### 2.2.2. Transparent Device

All the layers of the ST-PSCs were fabricated exactly as described for opaque PSCs until C60. After C60 deposition, the samples were loaded into the deposition chamber of an atomic layer deposition (ALD) tool, and 15 nm of SnO_2_ was deposited. Then, using shadow masks, a 140 nm indium-doped tin oxide (ITO) electrode was sputter-deposited. Finally, silver was selectively deposited via thermal evaporation for contact under a high-vacuum condition of <5 × 10^−6^ Torr.

#### 2.2.3. Solar Cell Characterization

The current–voltage (J–V) characteristics were recorded under simulated AM 1.5G irradiation (100 mWcm^−2^) produced using a class AAA solar simulator (Wacom, Tokyo, Japan). The AM 1.5G irradiation was calibrated with a standard Si cell (Newport, KG5 window). A metal mask with an area of 0.0942 cm^2^ (confirmed with microscope) was used to define the aperture areas of the devices. The scan rate was 100 mV/s.

### 2.3. Device Simulation

Device simulation was performed using the SCAPS-1D simulation software (ver. 3.3.08). The thickness of each functional layer was obtained via scanning electron microscopy analysis of the real devices, as shown in Figure 1.

Table 1 summarizes the input parameters used to establish an opaque device model based on the measured J-V curves of the real opaque device, as will be shown in the Results and Discussion section. The parameters not mentioned in Table 1 were set to be fixed at the following values obtained from the literature [20,21]: the effective density of the states of the conduction band and valence band were 2.2 × 10^18^ and 1.8 × 10^19^ cm^−3^, respectively. The thermal velocity of the electron and hole was 10^7^ cm·s^−1^. The defective energy level was the center of the band gap, and the defect type was neutral. The energy distribution was Gaussian, and its characteristic energy was 0.1 eV. The capture cross-section of the electron and hole was 2 × 10^−14^ cm^2^. Regarding the absorption of each layer, the pre-factor Aα was set to be 10^5^ cm^−1^ eV^−1/2^ to obtain the absorption coefficient (α) curve, which was calculated as α = A_α_(hv − Eg)^1/2^, where hv is the photon energy. In addition, the effects of the BCP layer; the series resistances caused by the substrate FTO/HTL and the electrode materials; and the reflection of each layer were not considered in this simulation for simplicity, as well as to show the effects of the C60/SnO_2_ interface more clearly, which will be described in more detail in the Results and Discussion section.

Regarding the simulation of the transparent device, a SnO_2_ layer was added to the opaque device model, with the light entering from the ETL side. The input parameters for the SnO_2_ layer are shown in Table 2. The conduction band position and carrier concentration were set to vary, while the other parameters were fixed. Furthermore, to gain further insight into the carrier transport characteristics at the SnO_2_/C60 interface, which can be affected by a combination of electrical properties of both layers rather than be determined simply by a single layer, the doping density of the C60 layer was also allowed to vary in the simulation, as will be shown later. For simplicity, the interface of the SnO_2_/back contact was assumed to be in a flat band condition, and the recombination at the SnO_2_/C60 interface was assumed to be negligible.

## 3. Results and Discussion

### 3.1. Establishment of a Simulation Model for the Opaque Device Based on Real Experimental Data

In the first part of this study, we optimized our opaque device model by fitting it to the experimental results. The p-i-n perovskite solar cells used in this work had the following architecture: FTO/Me-4PACz/PVK/C60/BCP/Ag. The fabricated device exhibited an open-circuit voltage (V_OC_) of 1.12 V, a short-circuit current density (J_SC_) of 23.10 mA/cm^2^, and a fill factor (FF) of 81.60%, resulting in a PCE of 21.17% (Figure 2).

In the simulation, the substrate FTO was not considered because the FTO/Me-4PACz interface formed a tunneling junction or an ohmic contact so that a flat band condition would be selected for the (substrate) FTO/Me-4PACz interface, as described in the literature [42]. Moreover, because the role of the very thin BCP layer (5 nm) inserted in between the C60 and Ag electrodes in the real device was to reduce the interface recombination and ensure an ohmic contact at that interface [42,43,44], in the simulation, the BCP layer was omitted by treating the C60/back contact as a flat band condition. Consequently, the structure of the perovskite solar cells considered in our opaque simulation model was Me-4PACz (1 nm)/PVK (500 nm)/C60 (25 nm).

As mentioned above, as a hole-transporting layer, a self-assembled monolayer Me-4PACz ([4-(3,6-dimethyl-9H-carbazol-9-yl) butyl] phosphonic acid) was used. It is well known that there is an interplay between the surface recombination velocity (*SRV*) and the contact band alignment at the interface, which affects the resultant surface recombination characteristics. An unoptimized band alignment and a high *SRV* value can significantly increase surface recombination, thus reducing the V_OC_ [13]. In line with this view, it has been reported that Me-4PACz enables efficient passivation and hole extraction at the Me-4PACz/perovskite interface, thus reducing the interface recombination and increasing the V_OC_ [45]. In concrete, Me-4PACz/perovskite reportedly has a well-aligned band structure with a slightly positive valence band offset of 0.05 eV; therefore, the defect density at the interface in our simulation was set to a low value, 1.5 × 10^8^ cm^−2^, to obtain an *SRV* = 30 cm·s^−1^, which agreed with the experimental value reported in the literature.

Regarding the C60/PVK interface, a higher defect density value of 1 × 10^10^ cm^−2^, corresponding to an *SRV* of 2000 cm·s^−1^, was selected, as this interface is known to induce large V_OC_ losses due to the significant interface recombination in real devices [13,45]. This high *SRV* at C60/PVK in our simulation (as well as in real devices) can be explained by the absence of any passivation layer in the C60/PVK interface. It is worth noting that the insertion of an ultra-thin (≈1 nm) LiF interlayer between the perovskite and C60, a representative passivator, has been reported to significantly increase the device’s V_OC_ by reducing non-radiative recombination while retaining a high fill factor [46]. In devices with LiF passivation layers, a low *SRV* at the C60/PVK interface of 450 cm·s^−1^ was reported experimentally, whereas it can increase to a value as high as 5600 cm·s^−1^ without passivation layers [26,27]. This confirmed that our chosen value was well within the experimentally reported range.

As an absorber layer, a perovskite with mixed cations and mixed halides, Cs_0.05_FA_0.86_MA_0.09_Pb(I_0.91_Br_0.09_)_3_, was used [19]. Based on the proximity in composition between our perovskite and that reported in the literature, input parameters for the simulation of the perovskite layer, such as the conduction band position, diffusion length, and doping density, were mostly benchmarked from the experimental values reported in the literature [19].

The carrier type and its density in the absorber were n-type and 10^11^ cm^−3^, respectively, and this was calculated based on the work function and conduction band position (determined using the UV photoelectron spectroscopy measurement reported in [19]). Thus, the absorber was almost intrinsic. The defect density of the absorber, one of the most important parameters affecting the device performance, was set at N_t_ = 1.5 × 10^13^ cm^−3^ to obtain the diffusion lengths of the electron and hole (L_n_ and L_p_) of 1.3 μm. This value was chosen to be in the reasonable range based on experimental reports stating that non-passivated perovskite layers have diffusion lengths of approximately 1 μm (determined from TRPL decay measurements), whereas for passivated perovskite layers, they are approximately 2 μm. A carrier mobility of 2 cm^2^ V^−1^ s^−1^ was adopted from the experimental diffusion lengths combined with the Einstein relation [19].

The resultant simulated J–V curve of the opaque device, obtained using all the input parameters considered above, all of which were selected to be in the physically reasonable range reported in the literature, is presented in Figure 2, together with the experimental results. A band diagram of the opaque device, constructed with the input parameters before the contact, is also shown in Figure 3a. The simulation model exhibited a V_OC_ of 1.12 V, a J_SC_ of 23.03 mA/cm^2^, and an FF of 81.87%, which produced a PCE of 21.17%. The excellent match between the simulated and experimentally measured device parameters suggests that the input parameter set was, indeed, physically meaningful at first glance.

However, it should be noted that even though our opaque simulation model successfully reproduced a J–V curve of the real device, strictly speaking, we cannot exclude the case that it may work only for a single specific set of input parameters. To further confirm the general validity of the model (and input parameters), we intentionally changed the applied bias to check whether the corresponding simulation results would also show a physically reasonable and acceptable trend. To this end, we examined the recombination current density (*J_R_*) at different applied bias values (V) using the equation below [47]:$$J_{R}=R_{front}\;+R_{back}+R_{SRH}=\frac{SRV_{front}p_{front}n_{front}}{p_{front}+n_{front}}+\frac{SRV_{back}p_{back}n_{back}}{p_{back}+n_{back}}+\frac{pn}{\tau (p+n)}$$
where *τ*, *p*, *n*, and *SRV* are the carrier lifetime, hole density, electron density, and surface recombination velocities, respectively. When bias was applied, the hole density and electron density at each interface shown in Figure 3a varied depending on the change in the band alignment. Figure 3b shows the band alignments at selected bias conditions of 0.00 (short-circuit condition), 0.96 (maximum power point, MPP condition), 1.08, and 1.12 V (V_OC_ conditions) for the simulated device. The distribution of the electron and hole density in the structure at each bias condition is also depicted in Figure 3c. Because the *J_R_* is determined according to the electron and hole density, the changes in the electron and hole densities with bias, which are shown in Figure 3c, were expected to result in changes in the *J_R_* of the interface (*J_IR_*) and bulk (*J_BR_*). The corresponding *J_IR_* and *J_BR_*, as functions of the applied bias, were calculated and are shown in Figure 3d. As shown in Figure 3d, the *J_BR_* was significantly higher than the *J_IR_* when the applied bias was lower than 1.08 V, while interface recombination dominated when the applied bias was higher than 1.08 V. This indicates that the obtained V_OC_ (1.12 V) of the simulation model (as well as our real device) was mainly limited by the interface recombination. This result is reasonably consistent with the fact that we used a non-passivated perovskite absorber in the fabrication of the device, and it suggests that reducing interface recombination will increase the device’s V_OC_ and, thus, the efficiency, as reported in the literature [19].

### 3.2. Simulation Model for a Transparent Device Based on Real Experimental Data

As mentioned earlier, a transparent device was simulated with a SnO_2_ layer added to the opaque device model and the light entering from the ETL side. Thus, the model used for the transparent device had the following structure: Me-4PACz (1 nm)/PVK (500 nm)/C60 (25 nm)/SnO_2_ (15 nm). As the first step, similarly to the case of the opaque device, we attempted to fit our transparent simulation model to the real experimental data to ensure the validity of the model. It should be noted that for the consistency of both the experiment and simulation, a real transparent device was constructed using the same materials as the opaque one up to the C60 layer; thus, in our transparent device simulation, we used the same input parameters for the HTL/absorber/ETL, and only those for the SnO_2_ were set to vary. Figure 4 shows the J–V curves of both the real transparent device and the simulation model. Our fabricated transparent device exhibited a V_OC_ of 1.11 V, a J_SC_ of 22.08 mA/cm^2^, an FF of 80.00%, and a PCE of 19.62%, whereas the simulation model produced a V_OC_ of 1.12 V, a J_SC_ of 21.92 mA/cm^2^, an FF of 80.34%, and a PCE of 19.72%. The input parameters for SnO_2_ in this fitting are summarized in Table 2. A good match between the experimentally determined and simulated parameters confirms the validity of our model.

Recalling that the main motivation of this work was to investigate the effects of the electrical properties of SnO_2_ and C60 (and the corresponding C60/SnO_2_ interface) on the device performance, the next step was to vary the input parameters of SnO_2_ and C60 in the above simulation model within a reasonable range, as reported in the literature. This is based on the literature mentioning that the optoelectrical properties of SnO_2_, including the conduction band position (affinity), band gap, and doping density, can significantly vary depending on the ALD process conditions [30,34,38,48], and these variations can have considerable influences on the carrier transport characteristics at the C60/SnO_2_ interface. In addition, we investigated the effects of not only SnO_2_ but also C60 and its doping density in the following simulation due to the report that the conductivity of C60 can significantly change during ALD SnO_2_ deposition. It has been reported that transparent p-i-n devices generally undergo several annealing steps at 100 °C for at least 2 h during the SnO_2_ ALD process, which unintentionally dopes the underlying C60 due to the transport of iodine ions from the perovskite layer to the C60 layer [30]. The conductivity of C60 was reported to be enhanced by several orders of magnitude due to iodine ion doping, even resulting in heavily doped C60 [49,50,51,52], which means that there can be clear interplay between C60 and SnO_2_. In this regard, the following sections will consist of two consecutive simulations: we first investigate the effects of the conduction band position of SnO_2_ on the device performance using the input parameters determined above; then, the interplay of the doping concentration of both the SnO_2_ and C60 and its impact on the carrier transport will be simulated. Table 3 shows the doping density ranges of C60 and SnO_2_, as well as the conduction band positions of SnO_2_.

### 3.3. Effect of SnO_2_ Conduction Band Position

In this simulation, we fixed the doping density of C60 at 1 × 10^18^ cm^−3^ and that of SnO_2_ at 1 × 10^16^ cm^−3^. We also changed the conduction band position of SnO_2_ from 3.7 to 4.4 eV according to the range reported in the literature, as shown in Table 3. The other input parameters were the same as those used in the transparent device model, except the doping densities of the C60 and SnO_2_. It should be mentioned that even though the C60 doping density of 1 × 10^15^ cm^−3^ (Table 1) successfully worked to reproduce the J–V curves of both our opaque and transparent experimental devices, as shown in Figure 2 and Figure 4, respectively, the relatively low doping density of C60 in some cases induced simulation errors, especially when we used a deep conduction band position of SnO_2_ along with it. Thus, in the following simulations, where we varied the input parameters for SnO_2_, a C60 doping density of 1 × 10^18^ cm^−3^ was used to achieve the generality of the model, and this worked over a wide range of SnO_2_ conduction band positions. As Table 3 demonstrates, both numbers were, in fact, well within the experimentally reported range, and the validity of our model was not affected by this.

Figure 5 shows the simulated parameters of the transparent device as a function of the SnO_2_ EC (or CBO between the C60 and SnO_2_ layers). The detailed parameter values are summarized in Table 4. The band alignments (before the contact) for selected CBO values of +0.3, 0.0, and −0.4 eV are also shown in Figure 6 for better visualization. As shown in Figure 5a,b, the J_SC_ and V_OC_ values were almost constant regardless of the CBO, whereas the FF and PCE showed strong dependence on the CBO. Interestingly, the FF exhibited a high value ~ 84.9% at a CBO range of +0.3 to −0.1 eV, and then decreased when CBO further reduced from −0.1 to −0.4 eV. When the CBOs became more negative, a sharp decrease in the FFs was observed; the lowest FF value of 76.09% was observed at CBO = −0.4 eV. Because the PCE showed the same trend as the FF, we concluded that our transparent device with the SnO_2_ layer showed an inferior performance at a relatively high negative value of CBO, and vice versa. We also noted from the simulation results that those high efficiencies over 21.6% could be obtained over a wide range of positive CBOs.

To understand the results shown in Figure 5, we further analyzed the three different CBO cases (+0.3, 0.0, and −0.4 eV) by considering the energy band diagrams of the structures before and after equilibrium. The energy band diagrams before the contact are presented in Figure 6. If we look at the C60/SnO_2_ interface first, when the CBO = +0.3 eV (Figure 6a), the Fermi energy level of the SnO_2_ (−3.84 eV) was higher than that of the C60 (−4.02 eV), thus inducing an electron transfer from SnO_2_ to C60 to achieve thermal equilibration. Consequently, at equilibrium, the conduction band and valence bands of the C60 bent downward at the interface with SnO_2_, as shown in Figure 7c. This local downward band-bending caused electrons to be attracted to the C60/SnO_2_ interface, thus enhancing the electron extraction. In the case of CBO = 0.0 eV (Figure 7b), a higher Fermi level of C60 (−4.02 eV) compared to that of SnO_2_ (−4.14 eV) caused the electrons to transfer from the C60 to the SnO_2_ layer. At equilibrium, the conduction and valence bands of C60 bent upward at the C60/SnO_2_ interface, while those of SnO_2_ bent downward toward the junction, both of which moderately hindered the electron transport. At CBO = −0.4 eV (Figure 7c), a significant difference in the Fermi energy level of C60 (−4.02 eV) and SnO_2_ (−4.54 eV) caused the transfer of a large number of electrons from the C60 to SnO_2_, resulting in a severe upward bending of the band in the CB and VB of C60. The stiff slope of the upward band-bending in the conduction band of C60 near the C60/SnO_2_ interface should result in a strong impedance of electron extraction. The band alignment at the C60/SnO_2_ interface, simulated above, can adequately explain the FF changes depending on the CBO at first glance.

It is also noted that in all cases, in principle, the built-in potential across the perovskite layer was expected to be the same, because it was determined by the difference in the Fermi levels of the HTL and ETL, considering the intrinsic nature of our perovskite. However, the calculated built-in potential across the perovskite layer (Figure 7b) was revealed to differ in different CBO conditions. This is thought to originate from the thin (~25 nm) nature of the C60 layer, according to which the magnitude of the built-in potential across the perovskite layer is influenced not only by the Fermi level of HTL and ETL but also by the number of electrons transferred from the SnO_2_ layer. In the case of CBO = −0.4 eV, some electrons would also be expected to transfer from the C60 to SnO_2_, affecting the overall electron density of the C60 layer in a manner that lowers the difference in the Fermi levels of the HTL and ETL, finally reducing the built-in potential across the perovskite layer. On the other hand, this effect is relatively small and negligible in the cases of CBO = 0 and +0.3 eV, respectively, for which the built-in potential across the perovskite layer is similar. Thus, in addition to the band-bending at the C60/SnO_2_ interface (and corresponding electron transport barrier), the magnitude of built-in potential across the perovskite layer is also influenced by the conduction band position of SnO_2_ and plays a certain role in determining the carrier transport in the device. From these simulation results, considering both the electron transport barrier at the C60/SnO_2_ interface and the built-in potential across the perovskite layer, it can be suggested that the conduction band position of SnO_2_ should be chosen for a CBO between C60 and SnO_2_, so that it is not too negative to achieve a high PCE.

### 3.4. Effect of Doping Density of C60 and SnO_2_

The finding obtained in the previous section that a simple variation in the conduction band position of SnO_2_ has profound effects on the charge transport characteristics of our transparent device motivated us to further investigate the effects of changing the doping concentration of C60 and SnO_2_. This was intended to more actively change the difference in Fermi levels between two layers.

To this end, in this section, the device performance change is investigated as a function of the doping density of both C60 and SnO_2_. The condition of CBO = −0.4 eV was chosen for this simulation because the device’s FF and PCE were low in this condition, as shown in Figure 5 and Table 4. This raises the natural question of whether the device performance can be enhanced with proper control of the doping density of C60 and SnO_2_. At CBO = −0.4 eV, the doping density of C60 and that of SnO_2_ varied from 1 × 10^18^ to 1 × 10^20^ cm^−3^ and from 1 × 10^15^ to 1 × 10^20^ cm^−3^, respectively. Because the V_OC_ and J_SC_ were found to be practically independent of these doping density changes, which is similar to the results in Figure 5, only the FF and the corresponding PCE changes are summarized in Table 5. It was found that the device was expected to have the lowest FF (75.00%) and PCE (19.07%) when the doping densities of both the C60 (1 × 10^18^ cm^−3^) and SnO_2_ (1 × 10^15^ cm^−3^) were low. Interestingly, according to this condition, if the doping density of one of these two layers were to increase, the FF and PCE would significantly increase. For example, when the C60 had a doping density of 1 × 10^18^ cm^−3^, the FF increased from 75.00 to 83.08% because the doping density of the SnO_2_ increased from 1 × 10^15^ to 1 × 10^20^ cm^−3^ (PCE from 19.07 to 21.11%). In addition, when the SnO_2_ had a doping density of 1 × 10^15^ cm^−3^, the increase in the doping density of the C60 from 1 × 10^18^ to 1 × 10^20^ cm^−3^ resulted in an increase in the FF from 75.00 to 84.86% (PCE from 19.07 to 21.53%). The highest FF and PCE were obtained at a condition in which the doping densities of both layers were at their highest (FF = 84.88% and PCE = 21.53%). This result suggests that high doping densities of SnO_2_ and/or C60 can result in high device efficiency, even when the C60 and SnO_2_ layers have large negative CBOs. In other words, if we can properly control the doping density of at least one of these two layers, it becomes possible to achieve a high PCE in a wide range of experimental windows.

To understand the simulated results, we first analyzed the band alignment in the device structure as a function of the doping density of SnO_2_ at a fixed doping density of C60 of 1 × 10^18^ cm^−3^, in which case the FF and PCE showed monotonic increases as the doping density of SnO_2_ increased. The built-in potential across the perovskite layer is shown in Figure 8a, and a magnified view of the C60/SnO_2_ interface is shown in Figure 8b. As the doping density of SnO_2_ increased, the initial difference in Fermi levels between C60 and SnO_2_ before contact decreases, thus causing fewer electrons to move from the C60 to SnO_2_ so that a high built-in potential across perovskite would form (Figure 8a). On the other hand, the electron extraction barrier height at the C60/SnO_2_ interface was shown to be independent of the SnO_2_ doping density in this case, as shown in Figure 8b. Thus, when the C60 doping density was relatively low, such as at 1 × 10^18^ cm^−3^, changing the electron density of the SnO_2_ may have a major impact on determining the magnitude of the built-in potential across the perovskite layers and dominates the carrier transport characteristics of the device. We also noted that, if we were able to increase the doping density of SnO_2_ to the level of 1 × 10^20^ cm^−3^, even with CBO = −0.4 eV (which showed limited PCE of 19.07% at low SnO_2_ doping density as shown in Table 5), it would be possible to achieve a transparent device PCE of up to 21.11%.

As a next step, we analyzed the case in which the doping density of the SnO_2_ was fixed at a low value of 1 × 10^15^ cm^−3^ while the doping density of the C60 varied. The corresponding energy band diagrams from our simulation are shown in Figure 9. We found that the built-in potential over the perovskite layers significantly increased when the doping density increased from 1 × 10^18^ to 1 × 10^19^ cm^−3^ (Figure 9a), and a further increase to 1 × 10^20^ cm^−3^ induced a minor change. The difference in Fermi levels between the perovskite and C60 layers and its effect on the magnitude of the built-in potential across the perovskite, as a function of the C60 doping density, easily explains this result. In addition to this, differently from the case shown in Figure 8b, the band alignment at C60/SnO_2_ was also significantly affected by the change in the doping density of C60. When the doping density of the C60 increased, the depletion region in the C60 layer significantly decreased; at a high doping density of C60 of 1 × 10^20^ cm^−3^, the depletion region greatly shrinks, which may provide tunneling of the photogenerated electrons to tunnel toward the SnO_2_, further improving the FF by up to 84.86% and the PCE by up to 21.53% (Table 5). From these simulation results, the doping density of the C60 impacted not only the built-in potential across the perovskite layer but also the electron extraction characteristics near/at the C60/SnO_2_ interface. These combined effects facilitated the photogenerated electron transfer across the device, resulting in high FF and efficiency values.

From all these simulation works, which were established based on real experimental data, we found that the electrical properties of SnO_2_ and C60 indeed had significant effects on the performance of our transparent perovskite device. In all cases, the PCEs of the devices were mainly determined by the FFs, which could be explained by the mutual effects of both the built-in potential across the perovskite layer and the electron extraction characteristics near/at the C60/SnO_2_ interface. Our model suggests that increasing the doping density of at least one of the two layers of C60 and/or SnO_2_ would enable achieving a high PCE of transparent devices over 21% in a wide range of process conditions. If both of the doping densities were to approach the highest value of 1 × 10^20^ cm^−3^, this may result in a maximum PCE of 21.53% with an FF of 84.88% (from 19.62% of our real transparent device, Figure 4). Our work demonstrates a detailed understanding of the effects of the electrical properties of SnO_2_ and C60 on the carrier transport characteristics of p-i-n-structured semitransparent perovskite solar cells. It also provides important guidelines for the design of SnO_2_ and C60 process parameters for the development of high-efficiency transparent solar cells.

## 4. Conclusions

We successfully established simulation models for opaque and transparent perovskite solar cells based on real experimental data. These models were further utilized to investigate the effects of the electrical properties of SnO_2_ and C60, including the CBO at the interface and the doping densities of the two layers, on the device performance. Our simulation reveals that as the first choice, the CBO between the C60 and SnO_2_ should not be too negative in order to maintain a substantial built-in potential across the perovskite layer and a low electron barrier at the C60/SnO_2_ interface, thus facilitating electron transport. However, even in the inferior condition of a large, negative CBO, proper control over the doping densities of C60 and SnO_2_ can increase the PCE. In concrete, increasing the doping density of at least one of the two layers of C60 and/or SnO_2_ enables achieving a high PCE of transparent devices over 21%, which was possible by increasing the built-in potential across the perovskite layer and enhancing the electron extraction at the C60/SnO_2_ interface. These findings are expected to be helpful in designing SnO_2_ and C60 process parameters to further develop high-efficiency, transparent solar cells with better controllability and reproducibility.

## Figures and Tables

**Figure 1 nanomaterials-13-03091-f001:**
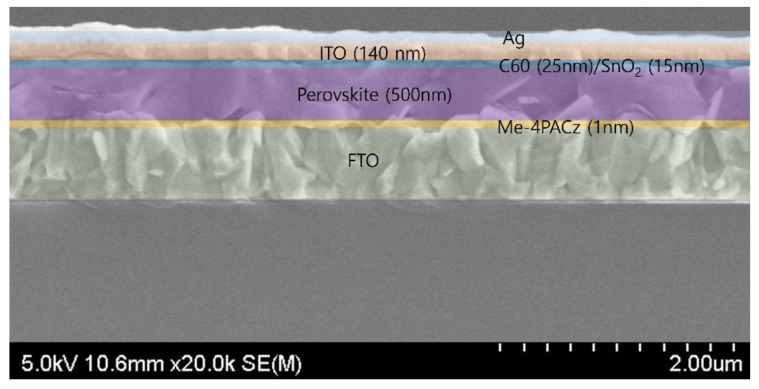
Cross-section SEM image of semitransparent perovskite solar cell.

**Figure 2 nanomaterials-13-03091-f002:**
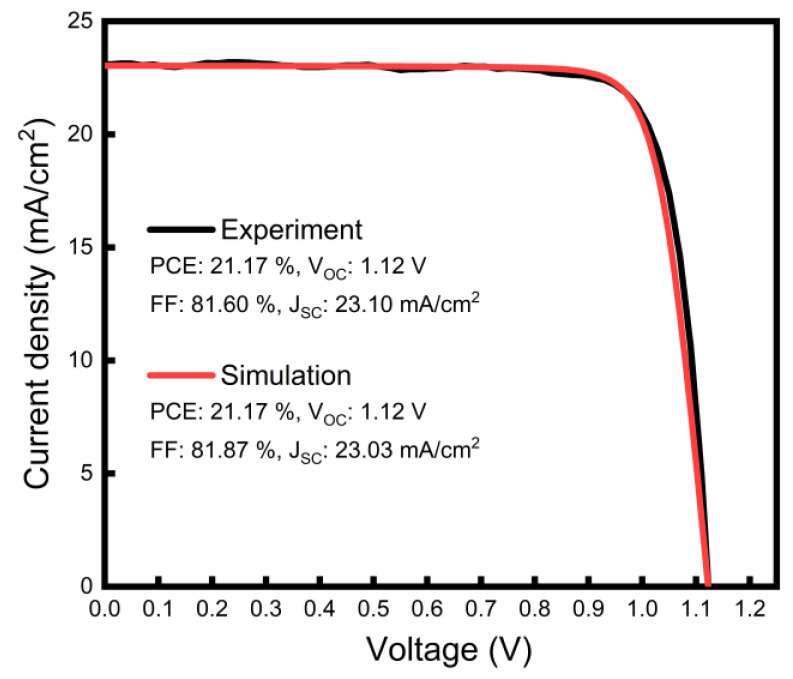
Experimental and simulated J–V characteristic of an opaque perovskite solar cell.

**Figure 3 nanomaterials-13-03091-f003:**
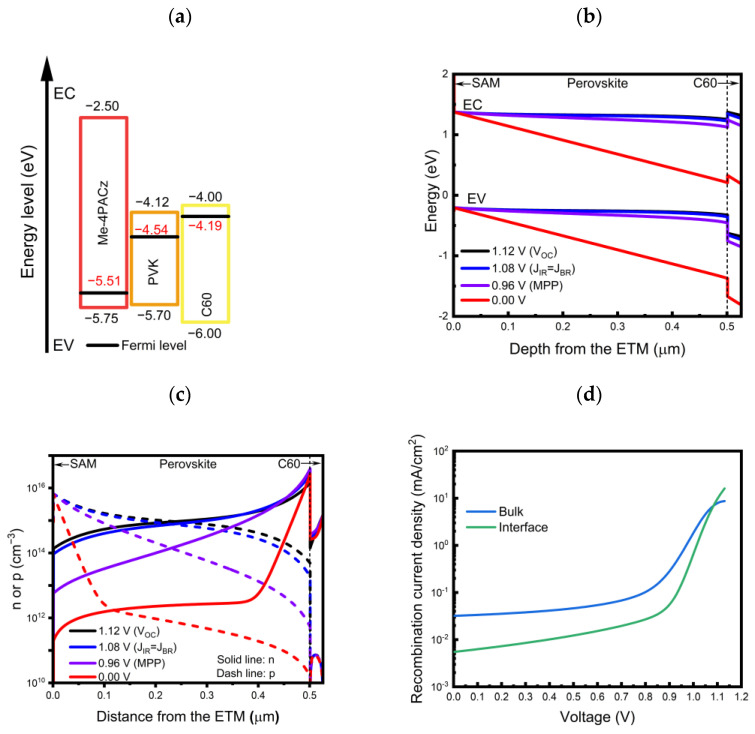
(**a**) The energy band diagram used for simulating the opaque perovskite solar cell before contact. (**b**) Band alignment in the opaque device structure at various applied bias values. (**c**) Electron and hole concentration at various applied bias values. (**d**) Interface and bulk recombination current density as a function of applied bias.

**Figure 4 nanomaterials-13-03091-f004:**
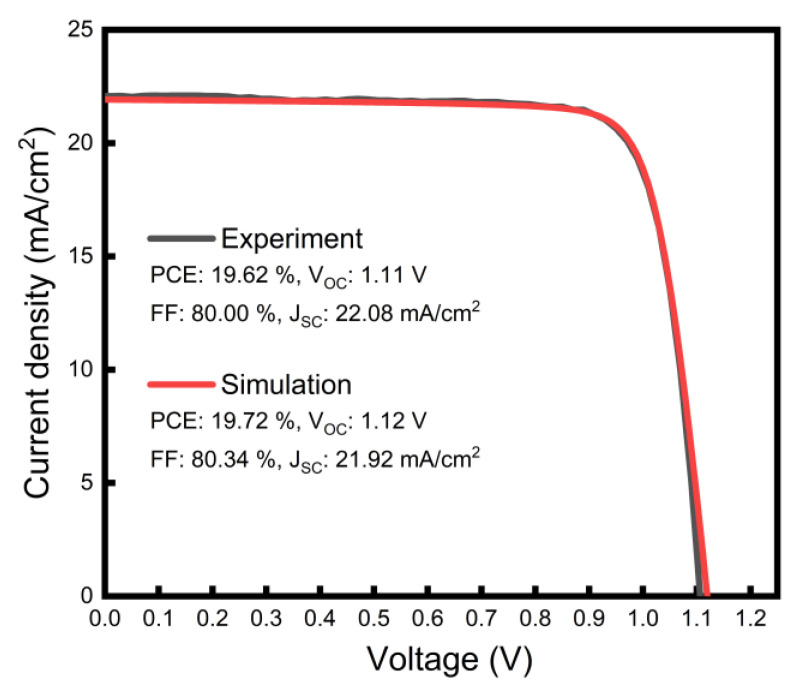
Simulated and experimental J–V characteristics of a semitransparent perovskite solar cell.

**Figure 5 nanomaterials-13-03091-f005:**
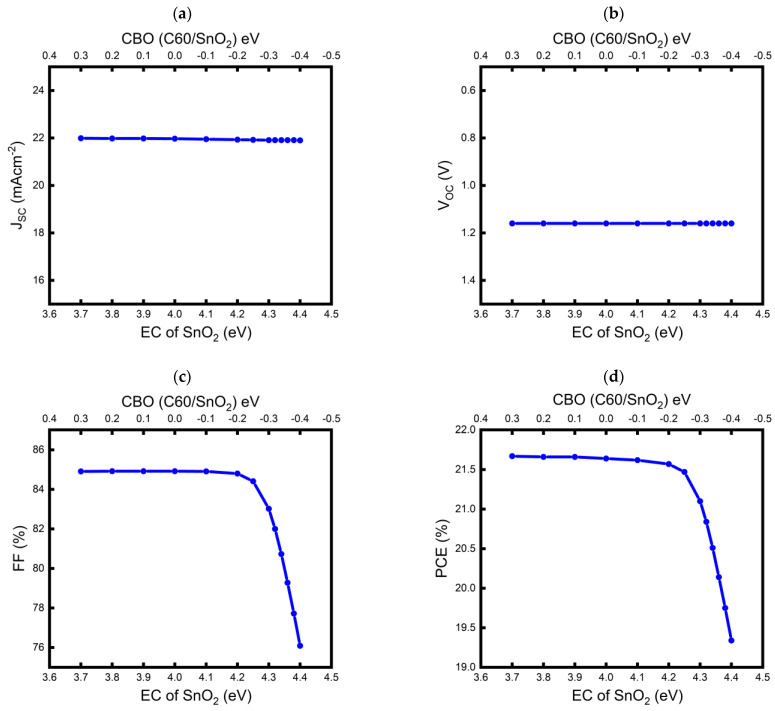
Device parameters of the semitransparent model as a function of a SnO_2_ conduction band, or conduction band offset CBO between C60 and SnO_2_: (**a**) J_SC_, (**b**) V_OC_, (**c**) FF, and (**d**) PCE.

**Figure 6 nanomaterials-13-03091-f006:**
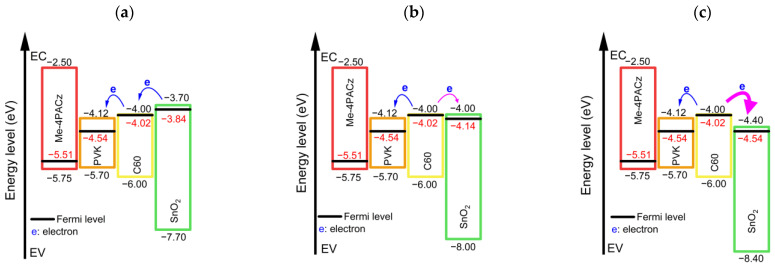
Energy band diagram (before contact) of the transparent model in various CBO conditions: (**a**) CBO = +0.3 eV, (**b**) CBO = +0.0 eV and (**c**) CBO = −0.4 eV.

**Figure 7 nanomaterials-13-03091-f007:**
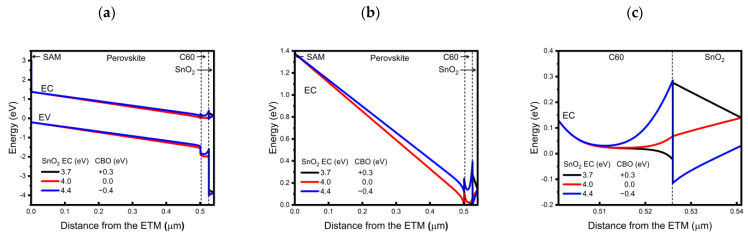
Simulated energy band alignment at equilibrium of the transparent model in various CBO conditions: (**a**) over the entire device structure; (**b**) magnified view of perovskite layer conduction band; and (**c**) magnified view near C60/SnO_2_ interface.

**Figure 8 nanomaterials-13-03091-f008:**
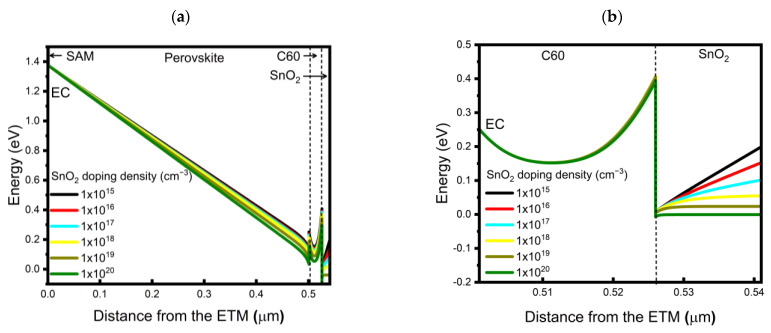
Simulated energy band diagrams as a function of a doping density of SnO_2_ at CBO = −0.4 eV and doping density of C60 of 1 × 10^18^ cm^−3^: (**a**) over the entire device structure and (**b**) near the C60/SnO_2_ interface.

**Figure 9 nanomaterials-13-03091-f009:**
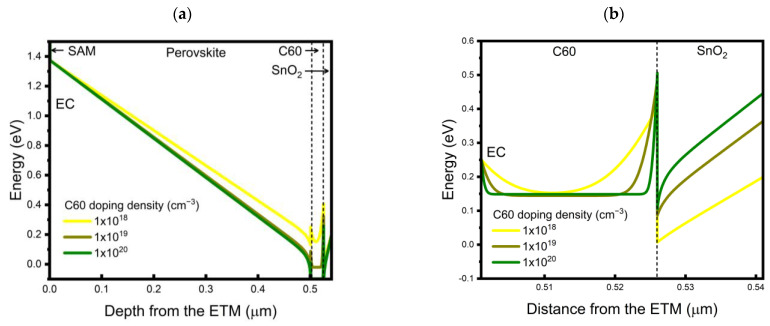
Simulated energy band diagrams as a function of doping density of C60, at CBO = −0.4 eV, and the doping density of SnO_2_ = 1 × 10^15^ cm^−3^: (**a**) over the entire device structure and (**b**) near the C60/SnO_2_ interface.

**Table 1 nanomaterials-13-03091-t001:** Input parameter of opaque device simulation.

Parameter	Me-4PACz	PVK	C60
Thickness, d (nm)	1 [22,23,24]	500	25
Band gap, E_g_ (eV)	3.25 [25]	1.58	2 [26,27]
Conduction band, EC (eV)	2.5 [13,25,28]	4.12 [19]	4.0 [26,29,30]
Relative permittivity, ε_r_ relative	10 [31]	10 [31]	5 [26,27]
Mobility of electron, µ_e_ (cm^2^/Vs)	6 [31]	2 [19]	0.01 [26,27]
Mobility of hole, µ_h_ (cm^2^/Vs)	24 [31]	2 [19]	0.01 (assume)
Donor density, N_D_ (cm^−3^)	0	1 × 10^11^ [19]	1.5 × 10^15^ (fit)
Acceptor density, N_A_ (cm^−3^)	1 × 10^15^ (fit)	0	0
Defect density, N_t_ (cm^−3^)	5 × 10^12^ [31]	1.5 × 10^13^ [19]	5 × 10^15^ [26,27]

**Table 2 nanomaterials-13-03091-t002:** Input parameter of semitransparent device simulation.

Parameter	SnO_2_
Thickness, d (nm)	15
Band gap, Eg (eV)	4.0 [32,33]
Conduction band, EC (eV)	4.1 [34,35]
Relative permittivity, ε_r_ relative	12.5 [36]
Mobility of electron, µ_e_ cm^2^/Vs)	20 [32,33,37]
Mobility of hole, µ_h_ (cm^2^/Vs)	20 (assume)
Donor density, N_D_ (cm^−3^)	1 × 10^17^ [30,32,35,38,39]
Acceptor density, N_A_ (cm^−3^)	0
Defect density, N_t_ (1/cm^3^)	5 × 10^15^ [36,40,41]

**Table 3 nanomaterials-13-03091-t003:** Ranges of conduction band position and doping density of C60 and SnO_2_.

Parameter	C60	SnO_2_
Conduction band, EC (eV)	4.0 [26,29,30]	3.7–4.4 [38,48]
Doping density, N_D_ (cm^−3^)	1 × 10^18^–1 × 10^20^ [30,49,50,51,52]	1 × 10^15^–1 × 10^20^ [30,32,33,37,48]

**Table 4 nanomaterials-13-03091-t004:** Solar cell parameters of semitransparent perovskite solar cells as a function of the SnO_2_ conduction band or the conduction band offset CBO between C60 and SnO_2_.

EC of SnO_2_	CBO	PCE (%)	V_OC_ (V)	J_SC_ (mA/cm^2^)	FF (%)
3.70	+0.30	21.67	1.16	21.99	84.91
3.80	+0.20	21.66	1.16	21.98	84.92
3.90	+0.10	21.66	1.16	21.98	84.92
4.00	0.00	21.64	1.16	21.97	84.92
4.10	−0.10	21.62	1.16	21.95	84.91
4.20	−0.20	21.57	1.16	21.93	84.80
4.25	−0.25	21.47	1.16	21.92	84.41
4.30	−0.30	21.10	1.16	21.91	83.02
4.32	−0.32	20.84	1.16	21.91	82.00
4.34	−0.34	20.51	1.16	21.91	80.73
4.36	−0.36	20.14	1.16	21.91	79.28
4.38	−0.38	19.75	1.16	21.91	77.72
4.40	−0.40	19.34	1.16	21.90	76.09

**Table 5 nanomaterials-13-03091-t005:** Changes in FF and PCE as a function of doping densities of C60 and SnO_2_ at fixed CBO = −0.4 eV.

FF (%)	Doping Density of SnO_2_ (cm^−3^)					
Doping density of C60 (cm^−3^)	1 × 10^15^	1 × 10^16^	1 × 10^17^	1 × 10^18^	1 × 10^19^	1 × 10^20^
1 × 10^18^	75.00	76.09	76.82	77.89	80.22	83.08
1 × 10^19^	83.42	83.46	83.50	83.58	83.91	84.50
1 × 10^20^	84.86	84.86	84.86	84.86	84.86	84.88
**PCE (%)**	**Doping Density of SnO_2_ (cm^−3^)**					
Doping density of C60 (cm^−3^)	1 × 10^15^	1 × 10^16^	1 × 10^17^	1 × 10^18^	1 × 10^19^	1 × 10^20^
1 × 10^18^	19.07	19.34	19.52	19.79	20.38	21.11
1 × 10^19^	21.21	21.22	21.23	21.25	21.33	21.48
1 × 10^20^	21.53	21.53	21.53	21.53	21.53	21.53

## Data Availability

Data are contained within the article.

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
