# Peer review of "Effects of the Electrical Properties of SnO2 and C60 on the Carrier Transport Characteristics of p-i-n-Structured Semitransparent Perovskite Solar Cells"

_nanomaterials, 2023, doi:10.3390/nano13243091_

Round 1

Reviewer 1 Report

Comments and Suggestions for Authors

The manuscript, titled 'Effects of Electrical Properties of SnO2 and C60 on Carrier Transport Characteristics of p-i-n Structured Semi-Transparent Perovskite Solar Cells for Tandem Application,' effectively addresses the phenomena associated with the performance of tandem solar cells. The analysis of different layers, along with theoretical studies, enhances the value of this work.

On the other hand, it is necessary to make some modifications that you could include in your manuscript to enhance its presentation.

To improve the manuscript:

-Clearly articulate the manuscript's contribution at the end of the text.

-Remove tables and abbreviations, as they are not appropriate.

-Eliminate color from the graphics and adhere to the design as it appears on the device.

-Include images from Atomic Force Microscopy or Scanning Electron Microscopy (SEM) illustrating the various layers, corresponding to Figures 5, 6, 7, and 8.

Comments on the Quality of English Language

Minor revision

Reviewer 2 Report

Comments and Suggestions for Authors

In this work, the authors investigated the properties of SnO2 and C60 and their influences on performances of transparent perovskite solar cells (PSCs). A simulation model was built successfully. Based on the simulation results, they predicted that increasing the doping density could benefit the performance of PSCs.

The manuscript is well constructed with sufficient data. It is acceptable for publication on the journal of Nanomaterials after addressing the following issues.

1.      More experiment data is suggested to be presented to confirm the prediction.

2.      In the introduction section, it is suggested to add more discussion about the background of interconnect layer in tandem solar cells.

3.      The following literatures are suggested to be cited.

Y. Huang et al., Nano Energy, 2021, 106219.

J. Zheng et al., Nature Energy, DOI: org/10.1038/s41560-023-01382-w

Reviewer 3 Report

Comments and Suggestions for Authors

In the article entitled “Effects of electrical properties of SnO2 and C60 on carrier transport characteristics of p-i-n structured semi-transparent 2 perovskite solar cells”, Pham et al. have employed simulation to study the electrical properties of SnO2 and C60 charge transport layers on the charge transport and device performance. Specifically, the authors pointed out the importance of the conduction band offset at SnO2/C60 interface towards better device performance. This is an important topic in the filed of perovskite solar cells and the understanding of which will facilitate further development of the field. I recommend its publication after the following points have been addressed.

1.     The authors said “The input carrier mobility of 1.6 cm2 V−1 s−1 is adopted from the experimental diffusion lengths combined the Einstein relation”. First of all, the reference is missing. And this number seems to be on the lower end of carrier mobility in this material. The authors need to do a complete survey of the literature to find a suitable value to use. 

2. The authors suggested that “the magnitude of the built-in potential across the perovskite layer is influenced not only by Fermi level of HTL and ETL but also by the amount of electrons transferred from SnO2 layer”. Does the model include the electron transfer efficiency and its contribution towards this? 

Reviewer 4 Report

Comments and Suggestions for Authors

The authors studied the effect of electrical properties of SnO2 and C60 on electron transport properties of opaque and semi-transparent perovskite solar cells. Combining simulation and experimental results, the authors presented and analysed the energy level alignment between SnO2 and C60 caused by doping The manuscript has been well organized and can be considered to publish after carefully checking some typos and grammars.  For example:

1.     Line 18 in Abstract. “has indeed significant impact” should be “has indeed significant impact on”.

2.     Line 188. “to be at fat-band condition.” should be “to be at flat-band condition.”

3.     The legend in Figure 4 c) should not be PCE but FF.

Comments on the Quality of English Language

The English does need to be improved.

Round 2

Reviewer 1 Report

Comments and Suggestions for Authors

Not comments